# Frequency of BRAF Mutations in Dysplastic Nevi, Lentigo Maligna, and Melanoma In Situ

**DOI:** 10.3390/jcm13164799

**Published:** 2024-08-15

**Authors:** Ivana Prkačin, Ivan Šamija, Nika Filipović, Matej Vucić, Majda Vučić, Nikola Ferara, Mirna Šitum

**Affiliations:** 1Department of Dermatovenereology, Sestre Milosrdnice University Hospital Center, HR-10000 Zagreb, Croatia; ivanaljubicicdr@gmail.com (I.P.); mirna.situm@kbcsm.hr (M.Š.); 2Department of Oncology and Nuclear Medicine, Sestre Milosrdnice University Hospital Center, HR-10000 Zagreb, Croatia; ivan.samija@kbcsm.hr; 3Department of Biology, Faculty of Science, University of Zagreb, HR-10000 Zagreb, Croatia; matej.vucic@biol.pmf.hr; 4Clinical Department of Pathology and Cytology Ljudevit Jurak, Sestre Milosrdnice University Hospital Center, HR-10000 Zagreb, Croatia; majda.vucic@kbcsm.hr; 5School of Dental Medicine, University of Zagreb, HR-10000 Zagreb, Croatia

**Keywords:** melanoma in situ, dysplastic nevi, lentigo maligna, BRAF, mutations

## Abstract

**Background:** In melanomas, mutations in the BRAF gene are common and their occurrence represents an early oncogenic event. Our goal was to determine and compare the frequency of BRAF gene mutations in dysplastic nevi (ND) and melanomas in situ (MIS), as well as whether there is a correlation between the presence of BRAF gene mutations and various anamnestic, clinical, and histopathologic variables. **Methods**: A total of 175 patients—106 with ND, 41 with MIS, and 28 with lentigo maligna (LM) were included in the study. DNA was extracted from tissue samples and analyzed using the competitive allele-specific TaqMan chain reaction by polymerase in real time to detect the presence of BRAF V600E and V600K mutations. The data were compared with anamnestic, clinical, and histopathological data. **Results**: There is a statistically significant correlation between the presence of BRAF mutation and the diagnosis of melanoma in situ (χ^2^ test, χ^2^ = 29.17, *p* < 0.0001). Patients with LM had a significantly lower incidence of BRAF mutations compared to patients with ND and MIS. There was a significant correlation between the presence of a BRAF mutation and tumor localization, as well as the age of the patient, but no statistically significant correlation between the presence of a BRAF mutation and sex, tumor size, or previous melanoma diagnosis. **Conclusions**: BRAF mutations in ND are essentially required; however, they are an insufficient oncogenic trigger for the development of melanoma. This research contributes to a better understanding of the etiopathogenesis of melanoma and the role of ND as possible precursor lesions.

## 1. Introduction

Malignant melanoma is a type of skin cancer that derives from melanocytes, skin cells that produce pigment [1]. In the last twenty years, melanoma has become one of the most frequent cancers in the Western world, with one of the fastest-growing incidence rates among all cancers [2]. Although melanoma accounts for only 4% of all skin tumors, it is responsible for 80% of skin cancer deaths. The five-year survival rate for patients with metastatic melanoma is only 14% [3]. Exposure to UV light is one of the most well-known environmental factors to cause melanoma [4,5,6], and intermittent or occasional intense exposure of parts of the body that are not normally exposed to sunlight and often result in burns with blisters is considered the most dangerous [3]. Despite the constant global increase of melanoma incidence, mortality rates are now growing at a slower pace [3]. Although patients with a genetic predisposition have an increased chance of developing melanoma, most cases are sporadic [7]. Sporadic cases account for approximately 90% of all melanoma cases worldwide, so the most significant causes of malignant melanocyte transformation are most likely environmental factors [8]. The risk of developing melanoma is also associated with a history of other malignancies in patients, especially malignancies of the brain, breast, prostate, and others such as sarcoma and lymphoma, as well as leukemia [9]. An increased number of nevi greatly increases the risk of melanoma, even if the nevi have a benign clinical and dermoscopic presentation [3]. Primary skin melanoma originates either from melanocytic nevi or de novo proliferation of malignant melanocytes [10]. Melanoma in situ is the abnormal growth of melanocytes in an unequal pattern, localized to the epidermis [10,11].

Dysplastic nevi are neoplasms with benign and malignant histopathologic features, making a clear-cut diagnosis difficult. They can be found on all parts of the body, somewhat more often on the head, neck, and chest. They begin to develop in adolescence and continue to develop even in the fourth decade of life [3]. Dysplastic nevus syndrome is often found in individuals who have a genetic predisposition for melanoma development. This syndrome is characterized by numerous dysplastic nevi and is associated with a risk of hereditary melanoma. They do not develop from an existing nevus but de novo [3]. Despite the morphological and biological similarity of dysplastic nevi and melanoma, malignant melanoma rarely develops from a dysplastic nevus [12]. Hence, a dysplastic nevus is considered an individual entity; it has the biological and morphological features of both a newly developed nevus and a melanoma. Dysplastic nevi are quite common in the European population (approximately 10%, spanning from 7 to 21%) [13].

Lentigo maligna is a type of melanoma in situ, a precursor lesion of lentigo maligna melanoma (LM) [14]. Lentigo maligna (LM) is often diagnosed in older people, mostly on sun-exposed parts of the body. It is usually not associated with melanocytic nevi. LM accounts for up to 4–15% of all invasive melanomas [15].

The genetic changes involved in the development of melanoma are complicated and not completely understood. Activation of the MAPK signaling pathway through mutations of the RAS and RAF genes plays a significant role [16,17]. Three RAF genes (ARAF, BRAF, and CRAF), which are part of the MAPK pathway, are found in all vertebrates [18]. In the 1980s, RAF was identified as an oncogene, and it is one of the key factors for cancer cell proliferation [19]. BRAF is the only RAF protein that is frequently mutated in cancer cells, and BRAF mutations are found not only in melanoma, but in various other carcinomas as well [19,20]. Still, the highest number of mutations is found in melanoma. Approximately 66% of melanomas have a BRAF gene mutation, predominantly a BRAF V600E-activating mutation, which accounts for up to 92% of BRAF mutations in melanoma [20,21]. In addition to cancer cells, nevi frequently harbor the BRAF V600E mutation, as evidenced by 82% of cases. Considering BRAF mutation is expressed in most melanocytic nevi, it is possible that the mutation is acquired in early stages of melanoma development [22,23].

The aim of our investigation was to determine and compare the frequency of BRAF V600 gene mutations in dysplastic nevi and melanoma in situ, as research on this topic is scarce and inconclusive. Furthermore, we wanted to determine whether there is a correlation between BRAF V600 mutation presence and various anamnestic, clinical, and histopathologic variables.

## 2. Materials and Methods

### 2.1. Tissue Samples

The patients included in this study were selected consecutively, and the key criterion for selection was that they had both melanoma in situ and one dysplastic nevus in their anamnesis. Testing was conducted on 175 formalin-fixed and paraffin-embedded tissue samples (Appendix A). They were confirmed with histopathological and immunohistochemical tests at the Ljudevit Jurak Pathology Department, Sestre Milosrdnice University Hospital Center, Zagreb. Patients’ ages ranged from 12 to 91 years at the time of sampling.

### 2.2. Tissue Dissection and DNA Extraction

After the pathologist defined parts of samples that corresponded to a dysplastic nevus or a melanoma in situ, microdissection was performed under microscopic visualization. For each sample, two to eight sections, 8 µm thick, were prepared and stored. Immediately after dissection, DNA was extracted using the QIAamp DNA FFPE Tissue kit (Qiagen, Hilden, Germany), following the manufacturer’s manual. Extraction yielded 80 µL of pure genomic DNA, and samples were stored at −20 °C.

### 2.3. Detection of BRAF Mutations Using Real-Time PCR

Testing for V600E and V600K BRAF mutations was performed on all DNA samples using mutation-specific TaqMan^®^ Mutation Detection Assay (Applied Biosystems, Foster City, CA, USA, SAD) in combination with the TaqMan^®^ Mutation Detection Reference Assay for the BRAF gene (Applied Biosystems). PCR reactions were prepared on a MicroAmp™ Optical 96-Well Reaction Plate (Applied Biosystems). For each reaction, 10 µL TaqMan^®^ Genotyping Master Mix-a (Applied Biosystems), 4 μL DNA (concetration: 6.25 μg/mL), 2 μL of the TaqMan^®^ Mutation Detection Assay or TaqMan^®^ Mutation Assay Reference Assay and 4 μL DI water was used. The total volume of reaction mix was 20 μL. Three reaction mixtures were prepared for each sample: (i) mix with Mutation Detection Assay for V600E mutation (BRAF_476_mu; Assay ID: Hs00000111_mu), (ii) mix with Mutation Detection Assay for V600K mutation (BRAF_473_mu; Assay ID: Hs00000769_mu), (iii) Reference Assay (BRAF_rf; Assay ID: Hs00000172_rf). A real-time PCR reaction was performed on an Applied Biosystems 9700 RT-PCR machine (Applied Biosystems) with the following conditions: initial denaturation at 95 °C for 10 min, followed by 5 cycles of denaturation at 92 °C for 15 s, and annealing/extension at 58 °C for 1 min, followed by 40 cycles of denaturation at 92 °C for 15 s and annealing/extension at 60 °C for 1 min. Each reaction was backed-up with a positive and negative control.

All reaction curves were analyzed in comparison to the threshold cycle value (CT0) of samples analyzed with Mutation Detection Assay for V600 BRAF mutations and Reference Assay for the BRAF gene to determine the presence of V600E or V600K BRAF gene mutations. To determine the CT value, the threshold was set to the value of 0.2. For each sample and each mutation, the ΔCT value was calculated with the following formula:ΔC_T_ = C_T(Mutation Detection Assay)_ − C_T (Reference Assay)_

Samples with ΔCT > 9.96 were negative for mutations, and samples with ΔCT ≤ 9.96 were positive for V600 BRAF mutations.

### 2.4. Data Analysis

Statistical analysis was performed using parametric or non-parametric statistical methods, depending on the distribution of the given data. In the case of the deviation of given data from the normal (Gauss) distribution, testing was conducted using non-parametric tests—Chi-square test (χ^2^), Fisher exact test, or Mann–Whitney U test. In the case of normal distribution, parametric tests were used. *p* < 0.05 value was considered significant. All statistical analyses were performed using Statistica v13.0. software.

## 3. Results

A total of 175 patients were included in the analysis, of whom 88 (50.3%) had a BRAF mutation. In patients with a dysplastic nevus (a total of 106 patients), 58.5% had a BRAF mutation. Of the 41 patients with melanoma in situ, 60.9% had a BRAF mutation, and in patients with lentigo maligna (LM), only one patient (3.6%) had a BRAF mutation detected (Table 1; Figure 1). There was a statistically significant correlation between a melanoma in situ diagnosis and the presence of a BRAF mutation (χ^2^ test, χ^2^ = 29.17, *p* < 0.0001).

BRAF mutations were detected in 44.7% of male patients and 58.3% of female patients. There was no statistically significant correlation between sex and BRAF mutation (χ^2^ test, χ^2^ = 2.646, *p* = 0.104).

The age of the patient was determined at the time of skin lesion excision. The median age of patients without a BRAF mutation was 61.5 years and ranged from 13 to 91 years. The median age of patients who harbored a BRAF mutation was 46 years and ranged from 12 to 81 years. Age at the time of excision was statistically significantly lower in patients with a BRAF mutation (Mann–Whitney U test, z = 3.227, *p* = 0.001) (Figure 2).

Statistical analysis showed a significant correlation between localization and the presence of the BRAF mutation (χ^2^ test, χ^2^ = 22.243, *p* = 0.0001). Regarding tumor localization, the highest number of BRAF mutations was found in patients with tumors on the trunk (61.5%) (Table 2). Additionally, 15.4% of patients with tumors on the head, 50% of patients with tumors on the upper extremities, and 27.8% of patients with tumors on the lower extremities harbored a BRAF mutation (Table 2).

No statistically significant correlation between the presence of a BRAF mutation and tumor size was found (Mann–Whitney U test, z = 0.834, *p* = 0.404) (Figure 3).

Finally, a comparison between the presence of the BRAF mutation and previous diagnoses of primary melanoma was also conducted. Out of 175 patients, 150 had no previous melanoma diagnosis, and 52% (78 patients) of these patients had a BRAF mutation. The BRAF mutation was detected in 40% of the 25 patients who had already been diagnosed with melanoma. There was no statistically significant correlation between previous diagnosis of melanoma and the presence of the BRAF mutation (χ^2^ test, χ^2^ = 0.267, *p* = 0.371) (Table 3).

## 4. Discussion

Malignant melanoma is one of the most aggressive human tumors. Although it accounts for only 4% of all skin tumors, it causes 80% of skin cancer deaths [3]. Dysplastic nevi are quite common in the European population; however, there is no universal clinical or histologic definition of this entity [13]. The presence of dysplastic nevi increases the risk of developing melanoma by 10 times, and most epidemiological studies have defined dysplastic nevi as a high-risk factor for developing melanoma [13,24]. Less than 1 dysplastic nevi in 200,000 cases turns into melanoma in both men and women aged 40 years or less. However, in men older than 60 years, the risk is higher, at 1 in 33,000 cases [25]. There is not a statistically significant difference in the number of BRAF mutations between people who have dysplastic nevi and people who have melanoma in situ (if lentigo maligna (LM) is taken out of the equation). These results support the role of BRAF mutations in dysplastic nevi as essentially required, but as an insufficient oncogenic trigger [26]. The presence of a BRAF mutation in a dysplastic nevus does not necessarily determine its transformation into melanoma, and it cannot be said with certainty if the dysplastic nevus will ever progress into any form of melanoma.

There was no statistically significant difference in the occurrence of BRAF mutations between men and women. These results are consistent with previous experience. The incidence rate of BRAF mutations does not differ between men and women since there is no difference in the occurrence of mutations, which are either genetically predisposed, mainly in younger patients, or affected by the environment (exposure to sunlight) in older patients [20].

Age plays an important role in the occurrence and presence of BRAF mutations in patients with melanoma. There was a statistically significant correlation between detection of BRAF mutations and the age of the patient; younger patients more frequently harbored BRAF mutations. Previous studies have also shown a considerable difference in the occurrence of BRAF mutations between people who are older and younger than 60 years [20]. An extremely high frequency of BRAF mutations has been reported in patients younger than 40 years of age, indicating a genetic predisposition for the occurrence of the mutation [20]. There is a possibility that the BRAF gene plays a role in melanocyte proliferation and differentiation in a growing and developing organism, and for that reason, mutations more often occur in younger patients [27]. A progressive decrease in the frequency of BRAF mutations was observed in older patients [28].

In this study, the primary tumor was located on the trunk of the body in 66.9% of cases. Previous research has shown that melanomas on the trunk have a peak incidence in the fifth and sixth decades of life, while melanomas of the head and neck occur in the eighth decade of life. In men, a higher incidence of tumors was reported on the back and shoulders, while in women, a higher incidence was reported on the lower extremities [3]. Given that the average age of the patients in this study ranged from 47 to 58 years, and a significant proportion of these patients were men, it is reasonable to anticipate a high proportion of patients with tumors located on the trunk of the body. The incidence of head and neck melanomas (14.9%) is also consistent with the existing literature (15 to 20%). Seeing that skin on the head and neck accounts for only 9% of the body’s surface area, this percentage is significant [29]. This result shows the importance of intermittent UV exposure in the pathogenesis of melanoma [3]. Head and neck melanomas generally have an unfavorable prognosis in comparison to melanomas of different anatomical localizations [30]. Our results show a higher incidence of BRAF mutations on the trunk of the body than on the head and neck, consistent with previously published results by Nikolaou et al. [31]. Head and neck melanomas differ from melanomas located on the trunk of the body as they are associated with chronic sun exposure, while melanomas of the trunk are intermittently exposed to UV light. Research has shown that melanomas associated with chronic sun exposure are genetically different from melanomas without such damage [32].

This study showed that there is a statistically significant difference in the frequency of BRAF mutations between patients with dysplastic nevi and those with melanoma in situ. Previous studies have shown disparate and inconclusive results [16,26,33]. However, excluding patients with lentigo maligna (LM) from the study revealed no statistically significant difference in the frequency of BRAF mutations between patients with dysplastic nevi and melanoma in situ. These results agree with the model that the BRAF mutation is an early event in melanoma progression and is associated with the development of melanocytic nevi. Considering that lentigo maligna (LM) is genetically different from melanoma that is linked to short-term sun exposure, BRAF mutations in LM are not as important. Our results contribute to elucidating the unclear role of dysplastic nevi as precursor lesions in melanoma development, favoring the model in which dysplastic nevi are separate entities and do not necessarily arise from ordinary nevi.

The results of this research are important for understanding the etiopathogenesis of melanoma and for future research in the field of melanoma development and the association of BRAF mutations with melanoma occurrence. The results of this study could have an impact on melanoma treatment. Selective BRAF and MEK inhibitors are the current standard treatments for metastatic melanoma, and the determination of BRAF mutations is a necessary step before treatment selection.

## Figures and Tables

**Figure 1 jcm-13-04799-f001:**
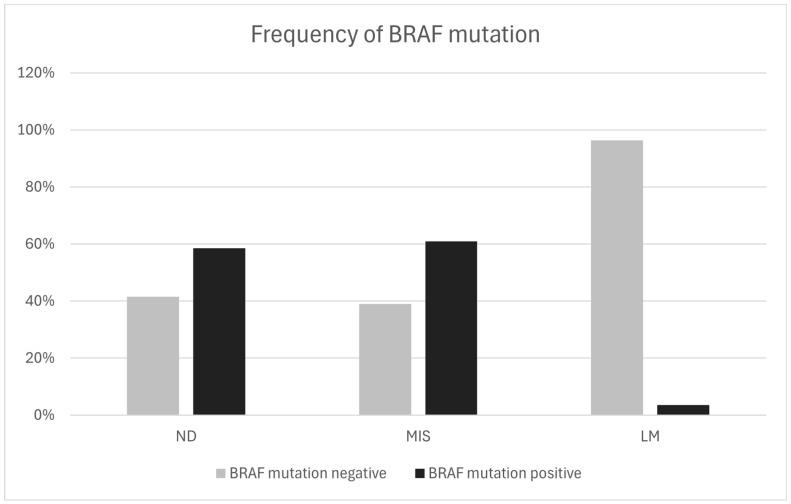
Presence and frequency of BRAF mutation between patients with ND, MIS, and LM.

**Figure 2 jcm-13-04799-f002:**
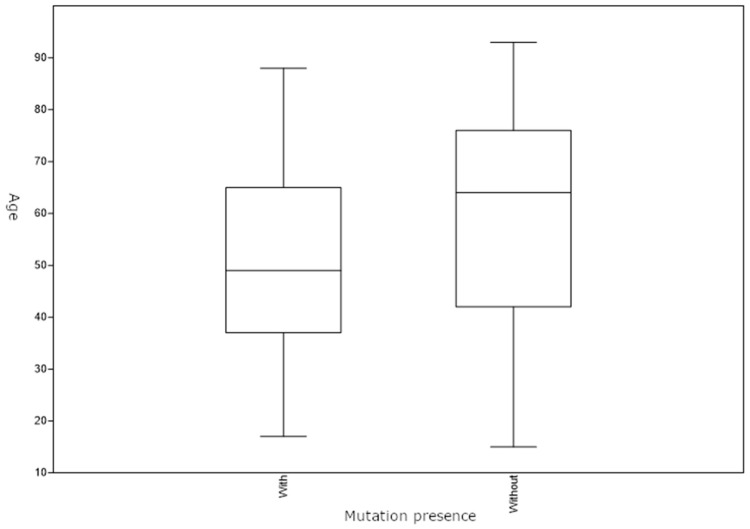
Age distribution of patients with and without BRAF mutation.

**Figure 3 jcm-13-04799-f003:**
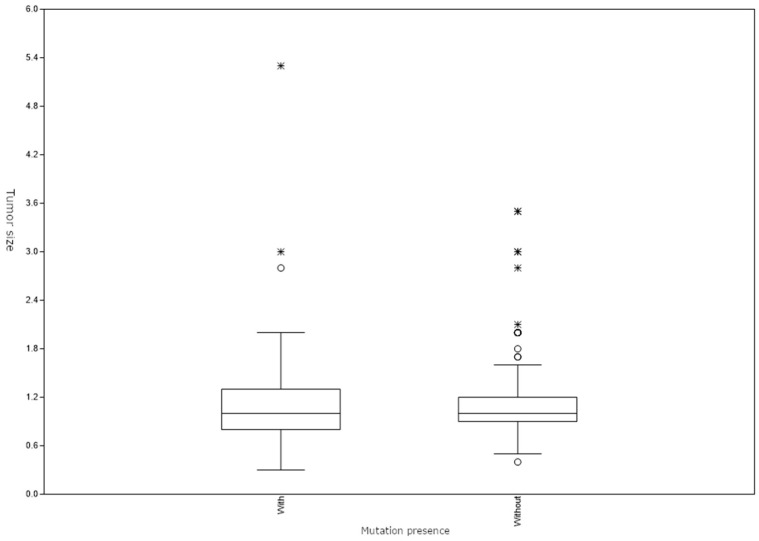
Tumor size distribution in patients with and without BRAF mutation. ∗ and ◦ represent outliers.

**Table 1 jcm-13-04799-t001:** The frequency of BRAF mutations varies across diagnoses.

Diagnosis	BRAF Mutation		Total
	No	Yes	
ND	44 (41.5%)	62 (58.5%)	106
MIS	16 (39.0%)	25 (60.9%)	41
LM	27 (96.4%)	1 (3.6%)	28
Total	87 (49.7%)	88 (50.3%)	175

ND—dysplastic nevus, MIS—melanoma in situ, LM—lentigo maligna.

**Table 2 jcm-13-04799-t002:** Correlation of BRAF mutation and localization.

Localization	BRAF Mutation		Total
	No	Yes	
Trunk	45 (38.5%)	72 (61.5%)	117
Head	22 (84.6%)	4 (15.4%)	26
Upper limbs	7 (50.0%)	7 (50%)	14
Lower limbs	13 (72.2%)	5 (27.8%)	18
Total	87 (49.7%)	88 (50.3%)	175

**Table 3 jcm-13-04799-t003:** Correlation between the presence of the BRAF mutation and a previous melanoma diagnosis.

Previous Diagnosis	BRAF Mutation		Total
	No	Yes	
No melanoma	72 (48.0%)	78 (52.0%)	150
With melanoma	15 (60.0%)	10 (40%)	25
Total	87 (49.7%)	88 (50.3%)	175

## Data Availability

Complete research data from this article can be found in a published version of the first author’s doctoral thesis in the Croatian national repository of dissertations and master’s theses. https://dr.nsk.hr/islandora/object/mef%3A6216/datastream/PDF/view, accessed on 2 July 2024.

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
