# Peer review of "Frequency of BRAF Mutations in Dysplastic Nevi, Lentigo Maligna, and Melanoma In Situ"

_jcm, 2024, doi:10.3390/jcm13164799_

Round 1

Reviewer 1 Report

Comments and Suggestions for Authors

The aim of this contribution is the comparison of frequencies of BRAF V600 gene mutations in dysplastic nevi and melanoma in situ.  Number of cases is not huge but high enough, and the molecular methods looking for BRAF mutation are correct and well described.

Results is divided in clear subheadings with concise descriptions of the experimental results. Interpretations are also correct. BRAF mutations are rare in LM, just one case(Table 1). Discussion emphasizes the results in comparison to other similar studies. Results are different of other reports. For instance, this study does not confirm the high frequency of BRAF mutations reported in patients younger than 40 years of age [20]. Comparison according to other factors, such as body part, are consistent. Research on this topic is inconclusive yet.

Only a minor point to be addressed. The abbreviations used for lentigo maligna are

LM, but also LMM. In some sentences, the abbreviation is not used, as line 147. I recommend the use of only one unified abbreviation .

Author Response

Comment 1: The aim of this contribution is the comparison of frequencies of BRAF V600 gene mutations in dysplastic nevi and melanoma in situ.  Number of cases is not huge but high enough, and the molecular methods looking for BRAF mutation are correct and well described.

Results is divided in clear subheadings with concise descriptions of the experimental results. Interpretations are also correct. BRAF mutations are rare in LM, just one case(Table 1). Discussion emphasizes the results in comparison to other similar studies. Results are different of other reports. For instance, this study does not confirm the high frequency of BRAF mutations reported in patients younger than 40 years of age [20]. Comparison according to other factors, such as body part, are consistent. Research on this topic is inconclusive yet.

Only a minor point to be addressed. The abbreviations used for lentigo maligna are LM, but also LMM. In some sentences, the abbreviation is not used, as line 147. I recommend the use of only one unified abbreviation .

Response 1: We would like to thank Reviewer 1 for his time and interest in reviewing our manuscript.

Abbreviations for lentigo maligna are corrected and unified throughout the text as suggested.

Reviewer 2 Report

Comments and Suggestions for Authors

The article ‘Frequency of BRAF mutations in dysplastic nevi, lentigo mligna and melanoma in situ’ may be an useful contribution to the journal; however, few changes should be taken into consideration, in the benefit of the reader:

Abstract: authors state ‘Our objective was to determine and compare the frequency of BRAF gene mutations in dysplastic nevi (ND) and melanomas in situ (MIS)’- for consistency reasons with title and the rest of the section, lentigo maligna should also be mentioned.

Line 130 h ΔCT ≥ 9,96 were negative for mutations, and samples with ΔCT ≤ 9,96 -the intervals seem to overlap; please clarify, in the interest of the reader

Lines 23-24 not clear what the authors mean by ‘when patients with LM were excluded from the analysis’, as the comparison between 2 groups per se does not need the presence or absence of a third party; this should be clarified.

Line 54 should be clarified ‘the upper layer of the skin’ with reference to the exact structures.

A table with patients demographics and descriptive statistics is mandatory; in the current form it is missing (age in various groups, sex, other relevant factors and so on…).

Materials and methods –lines 91-95: the authors should specify if the patients were consecutive and/or other inclusion or exclusio criteria were employed in patient selection.

Line 140 This section may be divided by subheadings.-not clear what is means that it may be devided.

A graph to display frequency of mutations between the 3 categories (just like Figure 1 bur for utation frequency) is mandatory; currently, it is missing.

Line 136 –authors state “-hi-square test”; is it Chi-square test (χ2)?

Lines 165-166 Statistical analysis showed significant correlation between localization and presence of BRAF mutation (χ2 test, χ2=22,243, p=0,0001).- the statement is not clear, specifically is not clear what topographical localisations are more frequent in patients with BRAF mutation>?

Lines 235-240 where authorst state ‘This study showed that there is a statistically significant difference in the frequency of BRAF mutations between patients with dysplastic nevi and those with melanoma in situ’ – it is confusing indeed, as in the Abstract the authors have stated ‘No statistically significant difference was found in the presence of BRAF mutations between patients with ND and patients with MIS’. It is confusing ot the reader and is a major issue, as it it the main  objective of the study. Therefore, it should be clarified. Inclusion and exclusion of LM is utterly confusing, also, and should be clarified, in the interest of the reader.

Grammar and punctuation must also be carefully checked within the entire article.

Comments on the Quality of English Language

Minor editing of English language required

Author Response

Comment 1: Abstract: authors state ‘Our objective was to determine and compare the frequency of BRAF gene mutations in dysplastic nevi (ND) and melanomas in situ (MIS)’- for consistency reasons with title and the rest of the section, lentigo maligna should also be mentioned.

Response 1: All co-authors agreed that the primary interest of this paper are BRAF gene mutations in dysplastic nevi (ND) and melanomas in situ (MIS) which should be kept in the spotlight and the results on LM will be discussed in our other paper.

Comment 2: Line 130 h ΔCT ≥ 9,96 were negative for mutations, and samples with ΔCT ≤ 9,96 -the intervals seem to overlap; please clarify, in the interest of the reader

Response 2: Corrected.

Comment 3: Lines 23-24 not clear what the authors mean by ‘when patients with LM were excluded from the analysis’, as the comparison between 2 groups per se does not need the presence or absence of a third party; this should be clarified.

Response 3: Whole sentence is removed since the wording in previous version was confusing and led to the wrong conclusion. Corrected as "There is a statistically significant correlation between the presence of BRAF mutation and the diagnosis of melanoma in situ (χ2 test, χ2=29.17, p<0.0001)."

Comment 4: Line 54 should be clarified ‘the upper layer of the skin’ with reference to the exact structures.

Response 4: Corrected.

Comment 5: A table with patients demographics and descriptive statistics is mandatory; in the current form it is missing (age in various groups, sex, other relevant factors and so on…).

Response 5: Added as a supplementary table.

Comment 6: Materials and methods –lines 91-95: the authors should specify if the patients were consecutive and/or other inclusion or exclusio criteria were employed in patient selection.

Response 6: Added.

Comment 7: Line 140 This section may be divided by subheadings.-not clear what is means that it may be devided.

Response 7: Typing error, text wasn't removed from the template document; corrected, text removed.

Comment 8: A graph to display frequency of mutations between the 3 categories (just like Figure 1 bur for utation frequency) is mandatory; currently, it is missing.

Response 8: Added.

Comment 9: Line 136 –authors state “-hi-square test”; is it Chi-square test (χ2)?

Response 9: Corrected.

Comment 10: Lines 165-166 Statistical analysis showed significant correlation between localization and presence of BRAF mutation (χ2 test, χ2=22,243, p=0,0001).- the statement is not clear, specifically it not clear what topographical localisations are more frequent in patients with BRAF mutation>?

Response 10: Previous sentence states: "Regarding tumor localization, the highest number of BRAF mutations was found in patients with tumor on the trunk (61,5%) (Table 2.). Additionally, 15,4% of patients with tumor on the head, 50% of patients with tumor on the upper extremities and 27,8% of patients with tumor on the lower extremities harbored a BRAF mutation (Table 2)." so we have now switched positions of these two sentences to make it more clear for reader. 

Comment 11: Lines 235-240 where authorst state ‘This study showed that there is a statistically significant difference in the frequency of BRAF mutations between patients with dysplastic nevi and those with melanoma in situ’ – it is confusing indeed, as in the Abstract the authors have stated, ‘No statistically significant difference was found in the presence of BRAF mutations between patients with ND and patients with MIS’. It is confusing ot the reader and is a major issue, as it it the main  objective of the study. Therefore, it should be clarified. Inclusion and exclusion of LM is utterly confusing, also, and should be clarified, in the interest of the reader.

Response 11: As we already stated in Response 3, this part is corrected in the abstract since the sentence in the abstract was constructed incorrectly. We agree that inclusion and exclusion of LM is confusing, so we removed that part, like the Reviewer advised in Comment 3: "as the comparison between 2 groups per se does not need the presence or absence of a third party".

Comment 12: Grammar and punctuation must also be carefully checked within the entire article.

Response 12: Corrected throughout the text.

Round 2

Reviewer 2 Report

Comments and Suggestions for Authors

The authors have addressed all suggested changes; in the current form, the material has been significantly improved.

Comments on the Quality of English Language

minor editing required